# Nanocellulose from Agricultural Wastes: Products and Applications—A Review

**Soledad Mateo, Silvia Peinado, Francisca Morillas-Gutiérrez, M. Dolores La Rubia** and **Alberto J. Moya** *

Department of Chemical, Environmental and Material Engineering, University of Jaén, 23071 Jaén, Spain; smateo@ujaen.es (S.M.); silviapeinadoserrano@gmail.com (S.P.); fmmg0008@red.ujaen.es (F.M.-G.); mdrubia@ujaen.es (M.D.L.R.)

* Correspondence: ajmoya@ujaen.es; Tel.: +34-953-212780

**Abstract:** The isolation of nanocellulose from different agricultural residues is becoming an important research field due to its versatile applications. This work collects different production processes, including conditioning steps, pretreatments, bleaching processes and finally purification for the production of nanocellulose in its main types of morphologies: cellulose nanofiber (CNF) and cellulose nanocrystal (CNC). This review highlights the importance of agricultural wastes in the production of nanocellulose in order to reduce environmental impact, use of fossil resources, guarantee sustainable economic growth and close the circle of resource use. Finally, the possible applications of the nanocellulose obtained as a new source of raw material in various industrial fields are discussed.

**Keywords:** lignocellulose biomass; nanocellulose; nanocrystal; nanofibers

## 1. Introduction

The social and environmental impact associated with the exploitation of fossil resources has resulted in the search for more environmentally friendly alternatives in recent years [1]. Lignocellulosic materials, one of the most important natural sources for the production of high added-value materials or biopolymers, are very attractive due to their biodegradability, low density and excellent mechanical properties [2,3]. Recent studies are focused on making use of wastes derived from agricultural activity due to its high content of lignocellulosic matter, composed in general by three fractions: cellulose, hemicellulose and lignin.

Cellulose is the most abundant organic substance in nature [4], an almost inexhaustible raw material and it is photosynthesized by green plants from carbon dioxide, water and sunlight. Cellulose can be considered as a renewable and biodegradable font of energy and raw material for other compounds' production as its simple but possesses an easily modified chemical structure, which can produce a wide range of fibres, films and functional polymers. The physical strength of cellulose is due to the fact that it is normally a crystalline polymer.

Lignocellulosic biomass (LCB) has been extensively studied and several research studies have shown the potential applications to be used as a starting substrate to produce value-added materials, such as biofuels and biopolymers [5–7]. Currently, the main sources of lignocellulosic wastes are divided into agricultural, forestry and industrial materials. Cellulose, as a main fraction of lignocellulosic biomass, is an unbranched natural polymer composed of repeating glucose units. Different treatments for the use of residual biomass (olive tree pruning, argan press cake, apple pomace and vine shoots or cotton, among others) have been applied for its transformation into nanocrystals and cellulose nanofibers.

Food packaging, biomedicine, mechanical reinforcement of matrices and membrane filtration among many other industries have used nanocellulose-based materials for their product applications [8]. Lately, nanocellulose has spurred research toward a wide range of products and applications, ranging from nanocomposites, gels and aerogels, viscosity

modifiers, films, barrier layers, fibers, foams, energy applications or filtering membranes [9]. The ordered structure of cellulose enables nanocelluloses to be isolated from LCB by employing processes that aim to disintegrate cellulose fibers into cellulose nanocrystals (CNC) and cellulose nanofiber (CNF), depending on the processing conditions [10].

Acid hydrolysis and mechanical treatments are the most widely employed methods for obtaining CNC and CNF, respectively; however, these procedures have some drawbacks related to economic and environmental aspects such as the energy demand of the process and the high amount of water required in the neutralization steps [11]. Another possibility for producing nanocellulose is the use of enzymatic hydrolysis, a promising environmentally friendly and sustainable route due to the advantages of enzymatic treatments, although the cost of enzymes may be an inconvenient. Some authors have evaluated the use of enzymes combined with mechanical treatments in order to facilitate fibrillation of cellulose bundles and to reduce energy requirements for nanocellulose isolation processes [12,13].

Nanocellulose is often used as a general term for different types of nano-sized and micro-sized cellulosic particles [14]. Nanocellulose is the disintegration product of cellulosic fibers to the diameter and length shown in Table 1, depending on the structure. Three types of nanocellulose morphologies can be distinguished: cellulose nanofibers (CNF), usually obtained by means of enzymatic and/or mechanical disintegration processes; cellulose nanocrystals (CNC), rod-shaped highly crystalline cellulose nanocrystals generally obtained by hydrolysis with concentrated mineral acids (mainly HCl or $H_2SO_4$); and bacterial nanocellulose (BNC), obtained almost exclusively by a family of bacteria known as *Gluconoacetobacter xylinius*.

**Table 1.** Nanocellulose products dimensions [15].

| Nanocellulose Product | Diameter, nm | Length, nm |
| --- | --- | --- |
| Cellulose nanofibers (CNF) Microfibril | 2–10 | >10,000 |
| Cellulose nanocrystals (CNC) | 2–20 | 100–600 |
| Bacterial nanocellulose (BNC) | 4–10 | 100–2000 |
| Cellulose nanofibers (CNF) Microfibrillated | 10–40 | >1000 |
| Microcrytalline cellulose (MCC) | >1000 | >1000 |

The main drawback for the production of cellulosic nanostructures is the great energy consumption derived from the high number of repetitive although necessary cycles for the degradation of the biomass structures. Pretreatment attempts to reduce this energy consumption and to obtain more efficient disintegration.

This review carries out a study of the main processes for obtaining cellulose nanofibers and nanocrystals as well as their optimization. Schematically, the process would consist of a series of stages: (a) conditioning of the biomass (milling, sieving and drying); (b) attainment of cellulose through pretreatments, mainly by alkaline processes, that degrade the structure of the lignocellulosic residue in order to facilitate the release of its monomers; (c) purification and bleaching; and (d) attainment of nanocrystals or nanofibers, mainly by acid hydrolysis.

## 2. Biomass Sources

Cell wall structure of LCB is composed mainly by three kinds of polymer: cellulose, hemicellulose and lignin, Figure 1. The proportion of these three components depends on the type and source of lignocellulosic biomass (different species).

Lignin represents approximately 5–35% by weight of dry lignocellulosic biomass, and its main function is to serve as a binder between and around the cellulose and hemicellulose fractions. This is what justifies the stiffness, compressive strength, decay resistance and water impermeability of the plant cell wall. Lignin is a cross-linked amorphous copolymer synthesized randomly from three different phenylpropane monomers: p-coumaryl, coniferyl and sinapyl alcohols. Depending on the species and source of lignocellulosic

biomass, the proportion of these three primary monomer units in lignin will be different. Recently, lignin has been used for the production of biofuels and chemicals from natural materials [16]. In addition, different lignin-based carbon materials have been used in catalysis processes, energy storage and pollutant removal.

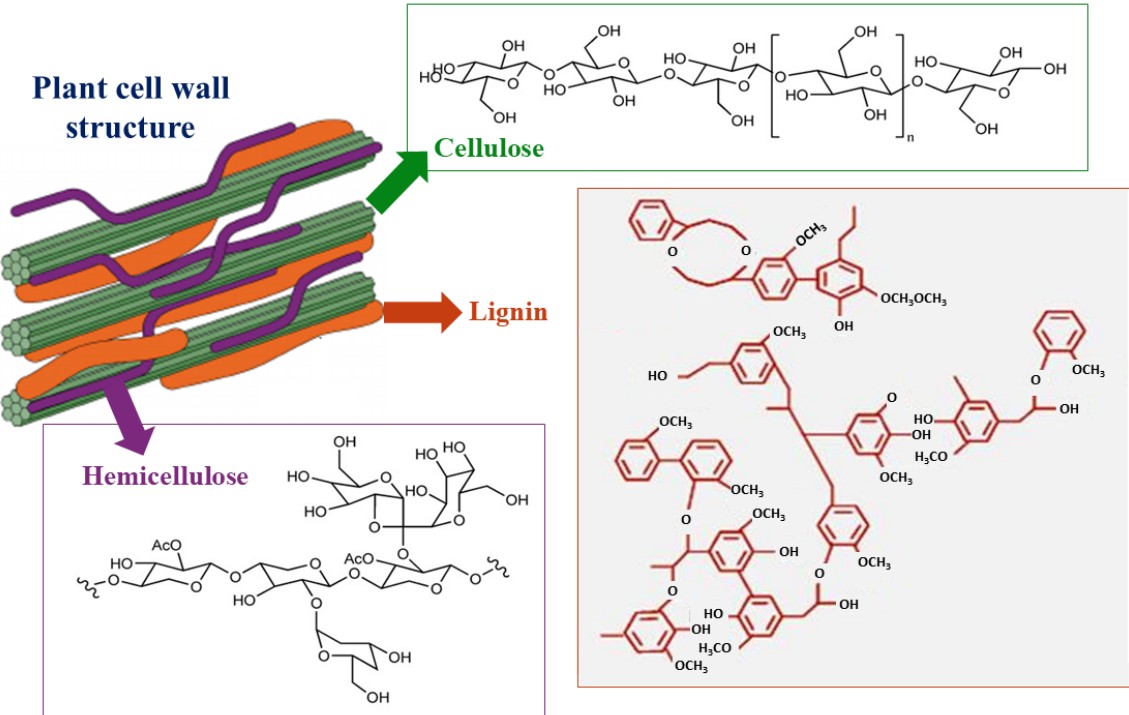

**Figure 1.** Scheme of the main components of lignocellulosic biomass.

Hemicellulose, 15–40% in lignocellulosic biomass, is an heteropolymer which is composed by short, linear and branched chains of different kinds of pentoses and hexoses as monomers. The most common types of hemicellulose are xylans and glucomannans. These are mainly found in the composition of hardwoods, while glucanans are mainly found in softwood [17]. Hemicellulose is attached to the cellulose fibrils through hydrogen bonds and van der Waal interactions. By using acid, alkaline or enzymatic hydrolysis in mild conditions of hemicellulose, ethanol and other valuable chemical products can be generated. Their applications in food, cosmetic and pharmaceutical industries possess great development.

The orientation of the glucose molecules and hydrogen bonding results in different cellulose allomorphs; the proportion of which depends fundamentally on the lignocellulosic biomass source and the treatment method. Generally, there are four types of cellulose allomorphs, namely cellulose types I, II, III and IV [14], Figure 2. Cellulose type I is the general allomorph of cellulose in nature or native cellulose, and there is a parallel packing of hydrogen-bond network. Cellulose type II comes from the chemical regeneration of type I by dissolving it in a solvent or swelling in acidic or alkaline solution. This fact justifies that in this allomorph the packing of the hydrogen bond network is antiparallel. Cellulose type III can be obtained from the ammonia treatment of cellulose I or II, while cellulose type IV is obtained by heating cellulose III up to 260 °C in glycerol.

Under the denomination of agricultural residue of lignocellulosic nature, all residues that are generated directly in the field are included. Depending on the crop, they can be grouped as woody crop residues that include the pruning of fruit, citrus, vine and olive trees; and herbaceous crop residues, which are formed by the remains after the harvest, where we would find the sunflower, corncob and the remains of cotton. Table 2 resumes the composition of some of the most usually employed wastes.

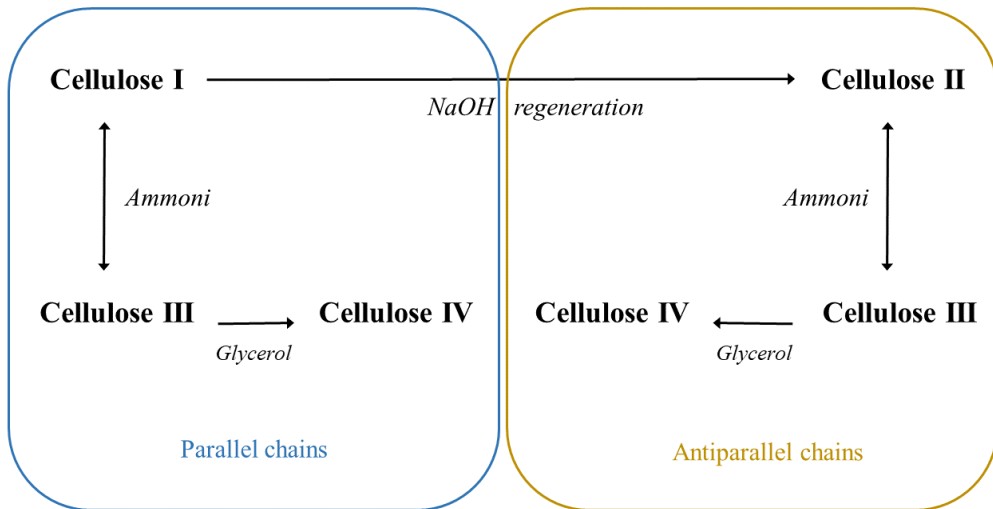

**Figure 2.** Polymorphs of cellulose and most the common ways to obtain them.

**Table 2.** Composition on the dry basis of the main agricultural wastes of lignocellulosic nature used.

|  | % Cellulose | % Hemicellulose | % Lignin |
|---|---|---|---|
| Aspen | 53 | 22 | 20 |
| Bamboo | 40 | 20 | 21 |
| Barley straw | 38 | 35 | 16 |
| Bean hulls | 52 | 26 | 10 |
| Coconut husk fiber | 25 | 12 | 40 |
| Corncob | 45 | 35 | 15 |
| Corn husks | 29 | 40 | 11 |
| Corn stover | 38–40 | 24–26 | 7–19 |
| Cotton | 40 | 23 | 23 |
| Jute | 61–71 | 14–20 | 12–13 |
| Mango seeds | 55 | 21 | 24 |
| Olive tree pruning | 30–40 | 24–27 | 18–23 |
| Pineapple leaf fibre | 75 | 13 | 10 |
| Pine | 47 | 20 | 27 |
| Pistachio shells | 54 | 20 | 25 |
| Poplar | 49 | 24 | 20 |
| Raw banana fibre | 70 | 20 | 6 |
| Rice straw | 30–35 | 25–35 | 12–23 |
| Sorghum bagasse | 34–45 | 18–27 | 14–21 |
| Soy hulls | 48 | 24 | 6 |
| Spruce | 43 | 30 | 28 |
| Sugarcane bagasse | 40–50 | 20–24 | 25–30 |
| Switchgrass | 5–20 | 30–50 | 10–40 |
| Sunflower | 37 | 21 | 17 |
| Wheat straw | 30–38 | 21–50 | 15–23 |

In addition to these residues, there are also pastes derived from the processes of obtaining argan oils (argan press cake) [18] or apple pomace [19].

## 3. Nanocellulose Products

Depending on the treatment carried out, two main types of nanocellulose structures, nanofibers or nanocrystals, are mainly obtained from the cellulose extracted in the pretreatments [20]. Most of the bibliography contemplates the extraction of cellulose and its

purification based mainly on acid treatments. In general, for obtaining cellulose nanocrystals after the alkaline treatment and the purification, an acid hydrolysis would be carried out, while the attainment of cellulose nanofibers is produced with mechanical treatments that culminate in a high pressure homogenization process. Moreover, bacterial nanocellulose can be obtained through the fermentation processes, see Figure 3.

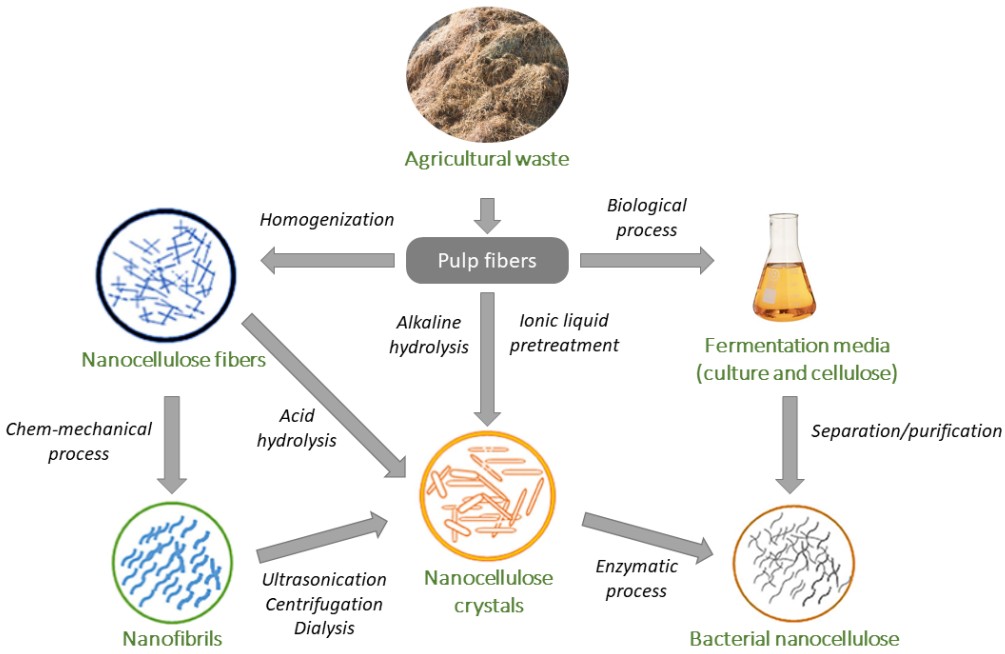

**Figure 3.** Schematic representation of nanocellulose production from agricultural wastes.

### 3.1. Microcrystalline Cellulose (MCC)

Microcrystalline cellulose (MCC) is a refined wood pulp product that can be used as a texturizer, anti-caking agent, emulsifier, extender and bulking agent in food production. In addition, it also has applications in vitamin supplements and as an alternative to carboxymethylcellulose. In the European Union, it is approved for use as a thickener, stabilizer or emulsifier, with the number E E460.

MCC also has applications in the cosmetic industry as an abrasive, absorbent, anti-caking agent, aqueous viscosity increasing agent, binder, bulking agent, emulsion stabilizer, slip modifier and texturizer, which can be found in various hair and skin care products as well as makeup.

The suitability for industrial use of MCC is based on different properties, namely particle size, moisture sorption capacity, moisture content, density, compressibility index, powder porosity, crystallinity index, hydration swelling capacity, crystallite size and mechanical properties such as tensile strength and hardness. Differential thermal analysis (DTA) or differential scanning calorimetry (DSC) and thermogravimetric analysis (TGA) are also widely used techniques for predicting the thermal behavior of MCC under thermal stresses. It is worth mentioning that the diameter of MCC fibers is widely determined by their source and the isolation process.

Acid hydrolysis is the main obtaining MCC process since the amorphous cellulose is degraded, leaving the MCC. The variables that most affect the hydrolytic process are the temperature and time of the hydrolysis process, the nature and concentration of the acid and the fiber-acid ratio. Thus, depending on these variables, MCC can be obtained with different particle size, crystallinity, morphology, thermal stability and mechanical properties, as well as different yields and degree of polymerization. However, among the main drawbacks that must be overcome, we must mention the cost, the amount of reagents, the dangers of corrosivity and the treatment of the effluents. Currently, research is still being carried out on the different acid hydrolysis processes in order to be able to isolate

MCC from different natural sources with applications in functional products with high added value.

The extraction of MCC by acid treatment has been compared to commercially available ones, showing good properties (thermal stability for example) [21].

Numerous methods have been recorded to isolate MCC from lignocellulosic wastes; thus, sodium chlorite followed by NaOH and acid hydrolysis have been used to obtain it from sisal fibers [22], by mercerization with 12% NaOH at room temperature for 2 h, acid hydrolysis and finally treated with NaOH [23] or using phosphotungstic acid to produce a new, green synthesis of MCC from waste cotton fabrics [24]. Finally, it has also been reported that MCC can be obtained by an enzymatic radiation process [25].

On the other hand, the steam explosion treatment process has been widely used as a promising mechanical pulping method [26]. Its main advantages compared to other technologies can be resumed in fewer hazardous chemicals, lower capital investment and environmental impact and more energy efficiency.

### 3.2. Cellulose Nanocrystals (CNC)

Cellulose nanocrystals are rod-like or whisker-shaped particles remaining after acid hydrolysis of lignocellulosic biomass. These particles have also been named nanocrystalline cellulose, cellulose whiskers and cellulose nanowhiskers, and they are nearly 100% cellulose.

The obtention includes an acid hydrolysis process necessary for dissolving the amorphous regions of the cellulose chains and for releasing the crystalline domains. In order to obtain CNC, it is required to subject the cellulose fibers to a purification process by means of chemical treatments in different stages. In general, an acid hydrolysis process involves the conversion of cellulose and hemicellulose polymers to their monomers. After this treatment, the cellulose that is released can be degraded to glucose (saccharification) or carry out another series of processes in order to take advantage of it. In hydrolysis with dilute acids, various acids such as formic, sulfurous, phosphoric, nitric, hydrochloric and sulfuric can be used, and the last two are the ones that have been used on an industrial scale due to their low cost and lower toxicity.

Acid hydrolysis can be performed by using different acids, being the most widely used hydrolytic agent sulfuric acid ($H_2SO_4$). The reason is that the esterification process of the anionic sulfate ester groups with the hydroxyl groups on the surface of cellulose induces the formation of a negative electrostatic layer on the surface of nanocrystals and facilitates their dispersion in water. Generally, the acid hydrolysis process occurs using 60–70% $H_2SO_4$ for 10–120 min, temperatures on the range 40–80 °C and using between 5 and 15 g of fiber per 100 $cm^3$ of acid disolution [27]. In the acid process, hydronium ions penetrate the cellulose chain hydrolyzing the glucosidic bonds of the amorphous regions and releasing the individual crystalline regions [28]. These nanocrystals have a short rod shape and are composed of 100% cellulose with high crystallinity (54–88%) [14,29], although their characteristics vary with the type of plant and extraction method.

After the hydrolysis process, deionized water is used to quench the reaction. This mixture is then subjected to a series of separation steps (centrifugation or filtration) and rinsing to remove the remaining acid or neutralized salt. A final separation or centrifugal filtration step can be used to remove any larger agglomerates in the final cellulose nanoparticle suspension. Ultrasonic treatments can be used to facilitate the dispersion of the crystalline cellulose in the suspension [27,29].

The morphological characteristics of CNC are mainly studied by using microscopic techniques such as atomic force microscopy (AFM), transmission electron microscopy (TEM), scanning electron microscopy (SEM) or light scattering techniques such as small angle neutron scattering (SANS), polarized and depolarized dynamic light scattering or dynamic light scattering (DLS) [30].

Among the various applications of cellulose nanocrystals, it is worth highlighting its application in the manufacture of biocomposite packaging materials [31,32].

### 3.3. Cellulose Nanofibers (CNF)

Cellulose nanofibers (CNF), which are also called microfibrillated cellulose, cellulose microfibril or microfibrillar cellulose, are currently being the subject of interest to researchers due to their interesting nanocomposite applications. In order to produce these structures, it is necessary to subject the cellulosic fibers to a mechanical disintegration treatment in order to cause the delamination of the fiber, thus isolating the nanometric fibers. This process requires high energy consumption in order to ensure the effectiveness of the nanofibrillation process. In order to reduce it and to increase the efficiency of the process, different pretreatments have been developed to undergo cellulose fiber before its nanofibrillation [33].

Obtaining glucose (monosaccharide) from cellulose can be achieved by various methods. One of the most promising is the use of cellulases, enzymes that catalyze the breakdown of cellulose polymer into smaller polymer branches, mainly cellobiose (dimer) and glucose. Traditionally, these enzymes are divided into three groups: endoglucanases, exoglucanases and cellobiohydrolases (CBH). The produced CNF (10–20 nm) show greater homogeneity compared to the ones obtained only with mechanical treatment [34].

The nanocellulose production method by enzymatic hydrolysis is less developed than alkaline or acid processes. However, this methodology has some characteristics that can be advantageous, so it is necessary to develop and invest production through enzymatic methods [11].

Partial carboxymethylation is a process in two consecutive stages where the hydroxyl groups of the cellulose chain are replaced by carboxymethyl groups ($CH_2COOH$) as a result of a chemical reaction between monochloroacetic acid and cellulose in the presence of sodium hydroxide. Carboxymethylation renders the fibrils' surface highly charged and facilitates their release during the disintegration treatment. Finally, nanofibers between 5–15 nm in diameter and more than 1 µm in length can be obtained [35].

Catalytic oxidation using 2,2,6,6-tetramethylpiperidine-1-oxyl (TEMPO) consists of using a selective oxidative catalyst that manages to negatively charge the surface of the fibers, causing the electrostatic repulsion of the anionic carboxyl groups between the oxidized fibers, to exceed the binding force. It is a widely used and is an effective pretreatment that reduces energy consumption up to 100 times during the mechanical disintegration process, and it results in a very homogeneous suspension of CNF between 3–4 nm and several micrometers in length; however, they are compounds of high cost [36,37].

The TEMPO reagent has resulted in nanofibrous structure products. If this process is combined with sonication action, the cellulose degradation is greater, becoming fibers characterized by their longitudinal breakage and making the next stages of the defibering process easier [38].

Mishra et al. [39] observed that the use of TEMPO agent in the treatment of oxidized cellulose pulp involved an increase in the degree of polymerization, nanocellulose yield and brightness stability due to post-oxidation.

### 3.4. Bacterial Nanocellulose (BNC)

Bacterial nanocellulose is another type of nanocellulose for which its main peculiarity, compared to the other types, lies in its production by bacteria through a process of assembling sugars of low molecular weigh (down-top processes). While cellulose nanocrystals and nanofibers are isolated from lignocellulosic biomass by using reverse methods, this nanocellulose is deconstructed of its cellulose fiber packaging [40].

The properties of BNC include, in particular, high purity (pure cellulose), include a nanofiber network structure and a high water content of 99% in the form of mechanically and thermally stable hydrogel bodies [41]. Scanning electron microscopy (SEM) is very suitable for studying the morphological properties of BNC since its single fibers have a thickness of tens of nanometers. The main advantage of this technique is that it does not require the preparation of ultra thin sections, as is required for transmission electron

microscopy (TEM). Additionally, its remarkably large depth of field is very useful for imaging three-dimensional BNC objects.

Although BNC has the same molecular formula of cellulose extracted from plants, it can be said that its characteristics compared to other types of nanocellulose are special. Depending on the objective that is intended to be achieved in the different applications, the bacterial nanocellulose can provide a resulting biomaterial with improvements in terms of mechanical qualities due to its biocompatibility, lack of toxicity, biofunctionality and ease of sterilization.

The need for high-quality BNC has triggered research in biotechnological techniques ranging from agitated to static cultivation approaches, using techniques ranging from batch or fed culture methods to continuous culture methods. As for the equipment used, these can include aerated fermenters, bubble columns and rotating disk reactors [42].

Depending on the application to be carried out, the choice of cultivation technique is essential since the structure and physical and mechanical properties of cellulose are influenced by the production method. Thus, the use of a specific strain of bacteria, the composition of the culture medium, the content of dissolved oxygen, the pH and the temperature of the culture medium is decisive in the productivity of the bioprocess. Among all these factors, the culture medium seems to be the most important one since it has a decisive influence on the total cost of BNC production; thus, identifying a low-cost culture medium that can improve BNC yield is essential in order to define an economically viable solution for industrial production oriented to application in a range of fields [40].

The excellent characteristics of the BNC material make it very versatile and applicable; therefore, it is expected that great developments in tissue engineering and regenerative medicine will occur in the future. However, the major challenges faced for the applications of bacterial cellulose are its up-scale production, the high cost of the media and the low productivity at industrial scale [43]. As mentioned, great advances are expected in many medical fields due to the application of BNC in medical devices; however, it should not be generalized but rather each specific application will have to be studied due to the great competitiveness of this field.

## 4. Conditioning, Pretreatments and Bleaching Processes

In order to extract and purify cellulose from LCB, a great variety of pretreatments and treatments processes have been assayed depending on the chemical nature of raw material. Figure 4 shows a general scheme that includes conditioning steps, pretreatments for cellulose obtention, bleaching processes and finally purification steps depending on desired nanocrystals or nanofibers.

### 4.1. Biomass Conditioning

The introduction in the processes for obtaining nanocellulose by pretreatment techniques basically pursues the rupture and fractionation of LCB components, which allows opening the structure of the material and facilitates subsequent processes [44–46].

An usual, the conditioning processes for breaking up cellulose into nanosize fibers involve grinding. The mechanism of fibrillation in the grinder is to break down the hydrogen bond and cell wall structure by shearing forces and individualization of pulp to nanoscale fibers. Some authors have produced nanofibers by passing bleached rice straw and bagasse pulps through a high shear grinder and homogenizer 30 and 10 times, respectively [47].

Another conditioning process applied to chemically treated and air dried biomass can be the use of liquid nitrogen in order to freeze them. Then, the biomass is grinded in a mortar, washed with distilled water and filtered. The cryocrushed fibers were soaked in 2 L water and mechanically defibrillated by using a disintegrator at 2000 rpm [48].

Another method for conditioning the biomass is high intensity ultrasonication (HIUS); it can be considered a mechanical process in which the hydrodynamic forces of ultrasound are used to isolate cellulose fibrils. During the process, cavitation generates a powerful

mechanical oscillating power and therefore high intensity waves, which causes microscopic gas bubbles that expand and implode, generating processes of disruption and the fractionation of the material [49].

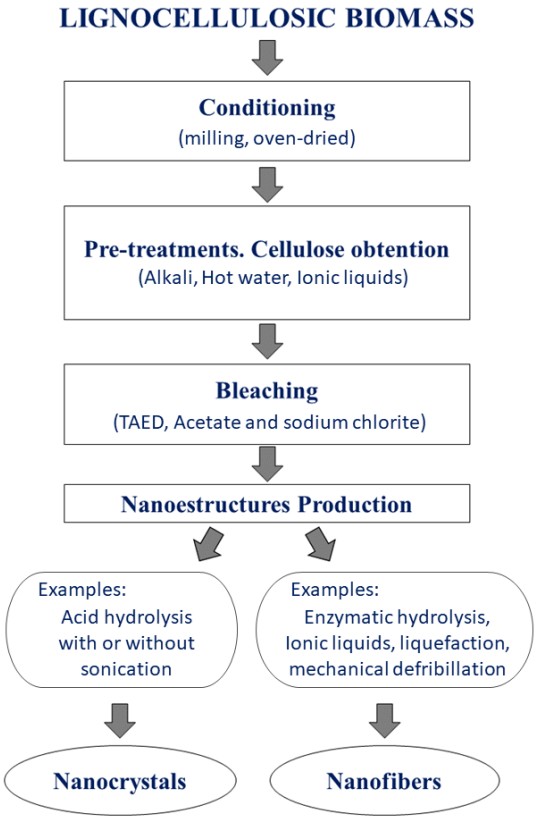

**Figure 4.** Scheme of the different types of processes for obtaining nanocrystals or nanofibers.

In most tests, before carrying out the pretreatment, the fibers must be cut into lengths and oven-dried at 60 °C [1,50]. The particle size of fiber-based nanocellulose was small, which had more advantages for future applications [4]. After pretreatment, the resultant fibers must be washed with distilled water until neutral pH, and then they were oven-dried at 70 °C or dried at room temperature until constant weight.

### 4.2. Pretreatments

#### 4.2.1. Alkaline

Alkaline is the most used pretreatment for obtaining cellulose for several reasons, among which include the ability to be carried out under mild conditions, possessing zero physical hazard, its low risk of inhalation danger, not being carcinogenic, not being persistent and it is non-bioaccumulative [51,52].

NaOH solutions cause the breakage of intermolecular $\alpha$- and $\beta$-aryl ether linkages between hemicellulose and lignin, causing swelling of the biomass pores, disruption of the lignin structure and an increase in available surface area [53–56].

The main disadvantages related to this pretreatment are the black generated hydrolyzate and processing time. These problems can be reduced by using this pretreatment in combination with others, which will result in lower concentration alkaline solutions being required and processing times reduced [49].

Depending on the type of biomass, different working conditions are established for the alkaline treatment in relation to NaOH concentration, time and temperature. Thus, the conditions for the extraction of crude cellulose from argan press-cake would be NaOH 12% and $Na_2SO_4$ 8%, with a temperature of 80 °C and for 90 min [18].

For a residue of the apple pomace, the parameters are optimized with NaOH concentration 10%, 70 °C and 160 min [19]. In the case of vine shoots, an NaOH concentration of 4% at 80 °C and 120 min is established [57]. For olive tree pruning, an of NaOH concentration of 1% at 60 °C and 60 min is optimized [58]. For corncob, a combined treatment is optimized in two stages, one with hot water and the other one with mild alkali using NaOH 2% at 90 °C for 90 min and a solid/liquid ratio ($w/w$) of 1/30 under magnetic stirring [1].

### 4.2.2. Hot Water and Ionic Liquids

Liquid hot water (LHW) and ionic liquids (IL) are among the main green pretreatment methods applied to fractionate biomass due to their fractionation efficiency and eco-friendly aspects.

LHW, also called autohydrolysis, is a chemical-free process that does not require the addition of a catalyst. In this pretreatment liquid water is used at high temperatures through the use of pressure. Thus, the formation of the typical degradation products that appear in acid processes (furans, phenolic compounds, aliphatic carboxylic acids or uronic acids) is low. On the contrary, the main disadvantage of this method is the high consumption of water and energy. Thus, LHW pretreatment is usually performed to enable enzymatic treatment by solubilizing hemicellulose. LHW is a pretreatment capable of substantially reducing the cell wall recalcitrance of lignocellulosic biomass, enhancing the saccharification of polysaccharides, particularly cellulose [59].

Ionic liquids are a group of salts made up of cations and anions that are liquid at room temperature. The special physicochemical properties of this compounds, among which the high solvation capacity to dissolve biopolymers, the adequate thermal and chemical stability and the low vapor pressure, have resulted in their increased use in pretreating LCB.

Among the ionic liquids, the ones that are most used for dissolving cellulose, whether it is refined or natural, include 1-butyl-3-methylimidazolium chloride (BmimCl), 1-allyl-3-methylimidazolium chloride (AmimCl) and 1-ethyl-3- methylimidazole acetate [60]. Recently, a lot of research studies are reporting the utilization of ionic liquids as potential solvents, swelling agents and catalysts for nanocellulose production.

The most important advantage of using IL as pretreatment is the minimal loss in solvent recovery, which can also be used many times. For example, it has been possible to recover more than 90% of BmimCl by reusing it four times without losing its activity [61]. Other authors have indicated that the use of IL as a nanocellulose surface modification medium may have applications in nanomedicine and drug delivery [62].

Finally, it should be noted that IL have recently been proposed as a reaction medium for homogeneous cellulose derivations. Thus, some authors have reported that homogeneous cellulose acetylation could be performed in AmimCl without any catalysts obtaining cellulose acetates with a wide range of degree of substitution [63]. On the other hand, it has been found that efficient O-acetylation of cellulose can be achieved by using a zinc-based ionic liquid [64].

### 4.3. Bleaching Process

The bleaching process, related to cellulose purification, is made in order to obtain an accurate removal of lignin after pretreatment. This step provides a more homogeneous end product. In most cases, bleaching is achieved by treating the fiber with chlorine in an acid medium.

There are different cellulose bleaching and purification techniques. After reviewing the literature in this study, we highlight the use of sodium acetate and chlorite [65] and the treatment with tetraacetylethylenediamine (TAED) [50]. However, it is necessary to mention that chlorine bleaching can be carried out in one or more stages. The final purification of the cellulose fibers obtained not only increases the removal of lignin but also reduces the diameter and improves some properties such as crystallinity or surface area. However, this procedure is not environmentally friendly and other more ecological but

perhaps less efficient bleaching processes should be applied, such as the use of ozone or hydrogen peroxide.

Acid-chlorite treatment is traditionally used in pulp industries, where it is called bleaching process. By combining sodium chlorite and acetic acid in aqueous medium, it can be remove most of the lignin from lignocellulosic biomass. The process temperature is 70–80 °C, and treatment time can vary between 4 and 12 h.

The use of acetate and sodium chlorite is a method applied on vine shoots, and it was based on performing a three times step consisting of the use of a mixture solution of equal parts (*v:v*) of acetate buffer (27 g of NaOH and 75 cm$^3$ of glacial acetic acid, diluted to 1 dm$^3$ of distilled water) and aqueous sodium chlorite (1.7% of NaClO$_2$ in water). This allows us to obtain pure white cellulose microfibers [65].

Corn husk was also twice treated by using 1.7% (*w/v*) sodium chlorite at 80 °C for 4 h, followed by cooling and filtering with distilled water [66].

Bleaching can also be performed by using oxygen and chlorine dioxide combined with hydrogen peroxide reinforced alkaline extraction stages. Elemental chlorine-free chlorine dioxide is generally formed by H$_2$SO$_4$ acidification of sodium chlorite and the collection of ClO$_2$ gas into cool water after passing through a sodium chlorite solution trap. The success of the process lies in achieving good delignification yields under mild conditions, avoiding degradation of the cellulose or the increase in costs [67].

Finally, the use of tetraacetylethylenediamine (TAED) has been carried out with a mild alkali solution on the fibers pretreated, after which they are bleached using 0.05% tetraacetylethylenediamine (based on dry fiber mass). The fiber suspensions would be placed in bags with double closure, immersing them in a water bath at 55 °C for 90 min. Subsequently, these fibers must be mixed manually every 10 min [50].

Table 3 resume the main pretreatment, treatments and products obtained from agricultural wastes.

**Table 3.** Sumary of pretreatment, treatments and products obtained from agricultural wastes.

| Biomass | Process | | Product | Ref. |
| | Pretreatment | Treatment | | |
|---|---|---|---|---|
| Apple pomace | NaOH 10%, 160 min, 70 °C | Acid hydrolysis 45% H$_2$SO$_4$, 50 °C, 45 min + ultrasonication | Cellulose nanocrystals | [19] |
| Argan press cake | NaOH 12% + NaSO$_4$ 8%, 90 min, 80 °C | | Purified cellulose powder | [18] |
| Bamboo | | Microwave liquefaction + NaClO$_2$ 0.1%, 75 °C, 1 h + ultrasonication | Cellulose nanofibers | [68] |
| Banana peels | KOH 5%, 14 h + NaClO$_2$ 1%, 70 °C, 1 h | Acid hydrolysis H$_2$SO$_4$ 1%, 80 °C, 2 h 3 times | Cellulose nanofibers | [69] |
| Citrulluss colocynthis | NaOH 1M, 24 h, 70 °C + bleaching with NaClO$_2$ | Acid hydrolysis H$_2$SO$_4$ 40%, 30 min + sonication | Cellulose nanofibers | [70] |
| Corncob | LHW 190 °C, 30 min, + Alkali 2% NaOH, 90 min, 90 °C | Ionic liquid 4% BmimCl | Cellulose nanofibers | [1] |
| Corn straw | NaOH 5%, 121 °C + H$_2$O$_2$ 2% + TAED 0.2% | Acid hydrolysis H$_2$SO$_4$ 64%, 25 °C, 90 min | Cellulose whiskers | [71] |
| Dunchi fibers | H$_2$O$_2$ 2% + TAED 0.2% | Acid hydrolysis H$_2$SO$_4$ 64%, 25 °C, 60 min | Cellulose nanofibers | [72] |
| Oil palm | NaOH 6%, 24 h, 20 °C | Acid hydrolysis H$_2$SO$_4$ 64%, | Cellulose | [73] |

**Table 3.** *Cont.*

| Biomass | Process | | Product | Ref. |
|---|---|---|---|---|
| | **Pretreatment** | **Treatment** | | |
| trunk | | 45 °C, 60 min | nanocrystals | |
| Olive tree pruning | | TEMPO-ox + defibrillation high-pressure microfluidizer | Cellulose nanofibers | [74] |
| Pine cones | NaOH 5%, 14 h | Acidified sodium chlorite 6%, 70 °C, 1 h | Cellulose nanofibers | [75] |
| Pineapple leaf | NaOH 2%, 4 h, 100 °C + bleaching 80 °C, 4 h | Acid hydrolysis $H_2SO_4$ 64%, 45 °C, 30 min | Cellulose nanocrystals | [76] |
| Rice husks | NaOH 4%, 2 h + bleaching 130 °C, 4 h | Acid hydrolysis $H_2SO_4$ 10 M, 50 °C, 40 min + sonication | Cellulose nanocrystals | [77] |
| Sisal fibers | NaOH 4%, 80 °C, 2 h + bleaching 80 °C, 4 h | Enzymatic hydrolysis (cellulases) | Microfibrillated cellulose | [78] |
| Sorghum stalks | NaOH + bleaching with $NaClO_2$ | Acid hydrolysis HCl 2.5 N, 105 °C, 15 min | Microcrystaline cellulose | [79] |
| Sunflower stalks | NaOH 5%, 2 h, 98 °C + sodium chlorite 5% | Acid hydrolysis $H_2SO_4$ 64%, 45 °C, 30 min | Cellulose nanofibers | [80] |
| Vine shoots | NaOH 4%, 120 min, 80 °C + bleaching with acetate and sodium chlorite | Acid hydrolysis $H_2SO_4$ 64%, 50 °C, 30 min | Cellulose nanocrystals | [65] |
| Yute fiber | NaOH 5%, 60 min, 80 °C + TAED | Mechanical defibrillation | Cellulose nanofibers | [50] |

## 5. Applications and Future Perspectives

Recent research studies are focused intensely on cellulose at the nano-level due to the consideration that it is an eco-friendly material, with special properties to be used as versatile structure with emerging applications. Main scientific fields based on cellulose nanomaterials attempt to take advantage of some of their valuable characteristics, such as excellent resistance, chemical inertness, renewability, transparent appearance, stiffness, biocompatibility, high specific area, biodegradability, low thermal expansion coefficient, stable structure, low density, high surface area-to-volume ratio and capacity for hydrogen bonding interactions [81,82].

Nanocomposites are nanomaterials that include nano-dimensional particles (discontinuous phase) into a reference matrix (continuous phase). In most cases, chains of different polymers are bonded by physical or chemical interactions (crosslinks) in order to produce multidimensional network structures such as hydrogels and aerogels [83]. The general applications of these polymeric-based structures encompass the following relevant areas: biomedical engineering, electronic, textile sector, energy, biomaterials and contamination removal systems that can be concreted considering the following practical applications, see Figure 5.

### 5.1. Food Packaging

One of the most studied area for this polysaccharide (nanocrystals or nanofibers) application is based on reinforcing synthetic polymers (elastomers, thermosets or thermoplastics) using small quantities of fillers (<5% by weight, [84]) to obtain lighter and more resistant-to-corrosion materials. Cellulose nanofibers with defined parameters have be used for the production of polymer composites with a strictly defined polymorphic structure, which may permit obtaining materials with good properties, enabling the thermoforming process of packaging [85,86]. The incorporation of nanocellulose into the polymeric matrix

is encouraged by weak (van der Waals forces and hydrogen bondings) and strong (covalent) intermolecular linkages [87], providing materials with suitable elasticity and more tensile strength.

Environmental care has promoted the replacement of conventional packaging using polyethylene or copolymer-based materials with biodegradable and renewable components [84], ensuring food quality and reducing wastes generation. In this manner, nanoclays, kaolinite or graphene (common fillers of plastic polymers) have been replaced with bio-based materials to improve active packaging consisting of antimicrobial nanoparticle addition to plastic films in order to prevent food contamination [88] or the use of layered composite films for more flexible packaging [89]. In this sense, nanocellulose is integrated with two types of polymers: thermoplastics and thermosets. Thermoplastics (polyethylene, polypropylene, polycarbonate, polyethylene terephthalate, polymethyl methacrylate, polyetherimide, polyamide, polyphenylene sulfide and liquid-crystal polymers) are capable of being molded after melting and ulterior solidification by cooling to retain their shape without chemical changes [90]. Integration of thermoplastics material and nanocellulose can only occur when the operation temperature is less than 300–350 °C [91]. On the other hand, thermosets materials (bismaleimides, epoxide, phenolic compounds, polyester, polyimide, polyurethane and silicone) are characterized for suffering irreversible liquid–solid transformations under specific chemical reactions. Cristalline nanocelullose has been successfully included in the films of both chitozan (polysaccharide derived from glucan units) and alginate, giving rise to effective interactions and improving the tensile strength [84]. On the other hand, packaging of carboxylated nanocellulose-based hydrogel from lignocellulosic materials has been employed as carrier of pH-sensitive dye for detecting freshness in food products [92].

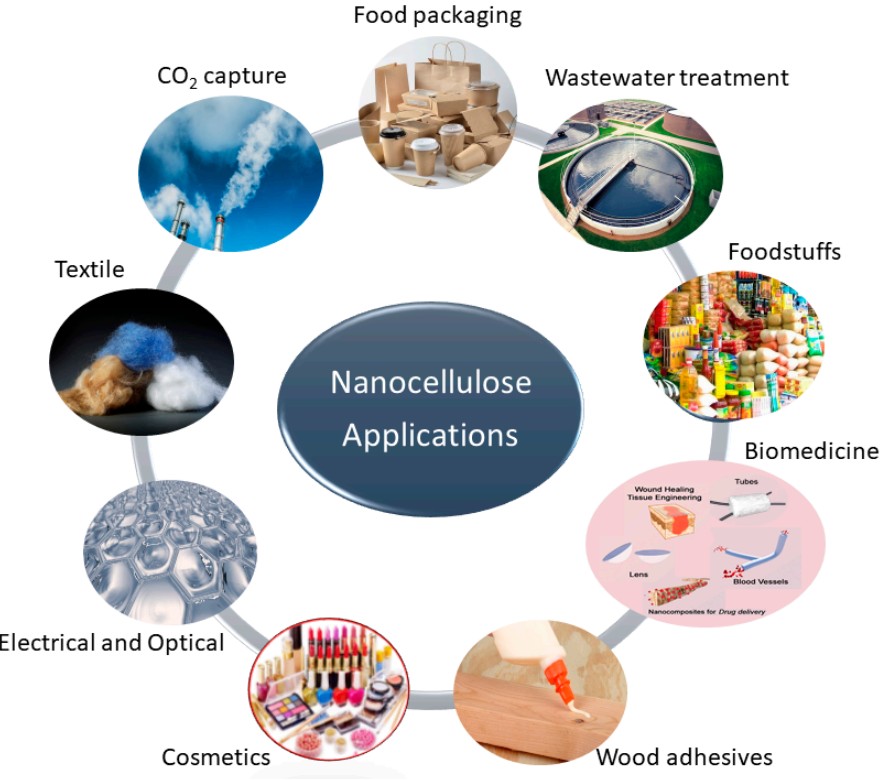

**Figure 5.** Main applications of nanocellulose.

Despite the enormous advances in studies about nanomaterials as reinforcements, supplementary research is needed in order to facilitate the inclusion of this nanostructure to the polimeric matrix; improve theoretical models to predict mechanical, chemical and thermal properties; and to reduce production costs for these nanocellulose-based struc-

tures. There is still a long way to go in terms of exploiting nanocellulose in commercial packaging. The employment of this biodegradable and sustainable material replacing its plastic counter-part will surely contribute the reduction in greenhouse gases [93].

### 5.2. Wastewater Treatment

The enormous amount of hydroxyl groups present on the nanocellulose surface allows establishing interactions with heavy metals and organic compounds such as methylene blue dye using green aerogel composed by cellulose nanofibril and graphene oxide-silica [94], Congo red dye with modified nanocellulose using polypirrolle [95] and disperse yellow dye [96]. At the same time, cellulose nanomaterials can act as filtering membranes to remove microorganism and additional pollutants from contaminated water [97,98]. Likewise, the purification of contaminated water can also be carried out by the addition of magnetic nanoparticles into cellulose polymeric matrix to separate metals such as $Ag^+$, $Cu^{2+}$ and $Hg^{2+}$, applying an external magnetic field [82,99]. Direct changes on the nanocellulose surface by sulfonation, amination and carboxylation could improve hazardous metal retention by adsorption from aqueous solutions, varying significant parameters as pH, adsorbent load, temperature and agitation speed, etc. In this sense, both humic and fulvic acids with negative environmental impact, can be removed by using nanocellulose modified by amine groups from wastewater considering an adsorption mechanism through electrostatic forces [100].

These nanostructures involve great advantages from an environmental perspective because of the feasible regeneration of this nanomaterial through desorption with strong acids and bases [101]. The nanocellulose use for pollutant adsorption possesses certain drawbacks due to situations of limited removal capacity for contaminants in aqueous solutions if comparing to the traditional adsorbents such as activated carbon, thus hindering its industrial application. This forces us, in many cases, to rely on chemical surface modifiers in order to increase the nanomaterial retention capacity, which causes higher production costs despite nanocellulose polyssacharides being considered as reusable materials. Currently, many companies worldwide produce huge quantities of nanocellulose from highly available lignocellulosic materials with affordable costs but advanced research studies in adsorption must be performed by using real wastewater and by using more effective adsorbents for scaling-up the laboratory assays.

### 5.3. Foodstuffs

Bacterial, microfibrillated, regenerated and vegetal nanocellulose has been added in food products as fat-replacer or oil emulsion stabilizer [102]. Furthermore, recent research is based on the implementation of this polymeric nanostructure to retain, by adsorption-desorption techniques, natural antioxidants such as polyphenol compounds that avoid food deterioration [103]. A jelly-like Ag decorated nanocellulose has been developed to detect hazardous residues in different food matrices (fish and pear), pesticides and veterinary drugs [104] by using surface enhanced Raman spectroscopy techniques.

The application of BNC obtained from agricultural wastes has been investigated in order to design packaging materials with components that release or absorb substances from or into the packaged food or the surrounding environment to extend the shelf-life and to maintain or improve the condition of packaged food [105].

Xie et al. [106] observed an improvement of the tensile strength and reduced water vapor and oxygen permeability and moisture content with the addition of BNC to potato peel-based films for food packaging applications.

### 5.4. Biomedical Applications

Nanocellulose-based composites are characterized for being highly biocompatible and scarcely toxic, which are the the major reasons for its use in numerous biomedical applications [107]. Incorporation of nanocellulose to drug delivery systems could control both the manner drugs are released and the interactions with target molecules, thus

increasing the effectiveness of drug administration. Changes on nanocellulose surface must be carried out to link drugs, non-ionic chemicals with hydrophobic character, to the nanopolymer. Mechanisms of sulfonation, silylation, esterification or carboxylation as well as cationic surfactants can be used to modify the negative charge of nanocellulose [20]; thus, enhancing its hydrophobic character to facilitate bonds with drugs can control its slow release according to recent research [108]. Hydrogels are capable of accommodating different biofactors that can be released by varying the physicochemical properties of the three dimensional network, for example, modifying the crosslinking density. For instance, two drugs (neocymin and diosgenin) can be supplied by incorporating gelatin cross-linked with genipin into diosgenin-modified nanocellulose, resulting in a hydrogel with great swelling capacity and antimicrobial activity [109].

Bacterial nanocellulose can also conform hydrogel tubes with multi-layered structure with the possibility of containing additives. Tubular nanocellulose can be used for cardiovascular implants as well as restorative material to repair areas of the bile duct, ureter and esophagus [110]. Some specific modifications of BNC have resulted in the production of compounds that can be substitutes for cartilage of the meniscus, auricular and nasal concha or even tubes for the regeneration of the nerves. Some specific modifications of BNC have resulted in the production of compounds that can be substitutes for cartilage of the meniscus, auricular and nasal concha or even tubes for the regeneration of the nerves. The obtained products are similar to natural tissues, with biocompatibility, moldability, biophysical and chemical properties fitting the needs of regenerative medicine [111].

Nanocellulose-based hydrogels have been applied in wound dressing applications due to some qualities of hydrogels related to antimicrobial and non-toxic properties and their additional capacity for absorbing toxin, maintaining the skin surface moist due to the porous hydrogel structure and accelerating wound healing [107]. In the same manner, the combination of nanocellulose with water or biological fluids results in hydrogels with properties quite similar to human tissues; thus, they are widely used in scientific areas such as tissue engineering and in vitro diagnostics [112]. Silica nanoparticles have been obtained from rice husks and were used to prepare BNC and silica composite aerogels [113].

### 5.5. Wood Adhesives

Synthetic formaldehyde-based adhesives such as phenol-formaldehyde resin, urea-formaldehyde resin and melamine-formaldehyde are the predominant adhesives in wood adhesive technology. However, the use of formaldehyde creates problems due to its emission and of other volatile organic compounds. For this reason, bio-based nanocellulosic materials have been developed to solve this disadvantage, helping sustainable development and also offering the benefit of their renewability [114].

Furthermore, cellulose nanomaterials in wood adhesives formulation significantly improves their adhesive bonding performance and stability in both wet and dry conditions and act as a reinforcing agent [115,116]. However, its content is limited by the strong increase in viscosity caused by its addition [117]. In conventional adhesives and, particularly, in the case of wood adhesives, nanocellulose finds application as it introduces a sustainable, abundant and cheap nano-biomaterial for property improvement [118,119].

Thus, the main advantages of using nanocellulose as a reinforcement for the production of reconstituted wood panels are summarized in the possibility of improving the mechanical and physical properties of the panels, altering the properties of the adhesives and reducing formaldehyde emissions. It has been observed that the addition of CNF in melamine-urea-formaldehyde and urea-formaldehyde adhesives reinforced the nature of cellulose nanofibers, improved energy and toughness fracture and enhanced the mechanical properties of the board.

### 5.6. Cosmetic

Nanocellulose has also been used and commercialized in the cosmetics industry. In particular, CNFs have been recently employed as an ingredient in mask packs and other

specific cosmetic assets. On the other hand, BNC encounters several potential applications in cosmetics due to its exceptional properties, mainly as a support for skin-active substances or as a structuring agent in cosmetic formulations [120].

Bacterial cellulose has also been applied as support for drug delivery, being effective for both hydrophobic and hydrophilic drugs [121]. Thus, a film made with bacterial cellulose produced in a modified medium has been successfully used to make sheet masks with a moisturizing and anti-inflammatory effect for skin prone to acne and inflammations.

The use of 1,3-dihydroxy-2-propanone in the formulation of compounds with microbial nanocellulose may be an alternative for patients who present vitiligo, as there are chances that this biomaterial will not cause allergies to the skin. This compound applied to the skin through bacterial cellulose also does not cause specific and unpleasant odor, which are typical effects of commercial cosmetics that contain it [122].

### 5.7. Electrical and Optical Materials

In order to achieve low-cost smart materials with biotechnological sensitivity, nanocellulose can be modified by using different surface groups, other polymer chains or even other nanoparticles; these processes allow obtaining materials with interesting properties and functions on their surface, specifically with electrochemical, electrical and optical applications [123].

The main objective may be the creation of chiral nematic compounds from cellulose nanocrystals together with other polymers or silica; this allows some component to be selectively eliminated later or used as a nanocomposite. Selective removal of a part of the starting nanocomposite provides chiral mesoporous material that can host materials or serve as a template to obtain other chiral materials [124].

On the other hand, in the application of lithium ion batteries, nanocellulose has been applied together with carbon nanotubes as current collectors, replacing aluminum foil and achieving improvements of around 17% [125].

Potential applications in the electronics field range from chiral plasmonics, sensitive hydrogels, anti-reflective coatings, optical filters, versatile electronics and soft actuators. On the other hand, cellulose nanocrystrals have been employed to develop optical materials, including surface plasmons, UV blocking and fluorescence and low refractive materials [30].

### 5.8. Textile

Due to the molecular structure and its large active surface, nanocellulosic materials have found great applications in the textile field, especially for medical applications. The special specific characteristics can be summed up in its antistatic behavior, low level of impurities and humidity and good mechanical and liquid adsorption properties [124].

Cellulose fibers are one of the most interesting materials for antimicrobial functionalization. Currently, the alteration of the cellulose fiber surface is considered one of the best techniques for achieving textil durability for medical use. For example, cellulose reacted with methylol-5, 5-dimethylhydantoin in combination with hypochlorite forms chloramines on the surface of the fiber, causing antimicrobial activity. Another process is the production of ethylcellulose nanocomposites with spirooxazine as a light stabilizer.

Cellulose nanoparticles preserve photochromic characteristics without affecting physical properties during the high temperature paste printing process [126].

### 5.9. $CO_2$ Capture

Two of the main strategies for $CO_2$ adsorption implies the nanocellulose chemical modification or inclusion of inorganic particles into nanocellulose; these nanomaterials, especially functionalized nanocelullose aerogels, can be employed as adsorbents or constituents of membranes to retain $CO_2$ selectively [127]. In this sense, nanocellulose has been employed to take part of physical (hydrogen bonds) or chemical (borax, epichlorohydrin or glutaraldehyde) cross-linked hydrogels that can be transformed to aerogels by freeze-drying cycles. Modifications of nanocellulose aerogels with silanes

such as 3-aminopropylmethyldiethoxysilane (3-silane derived) and N-(2-aminoethyl)-3-silane derived, promote $CO_2$ capture by chemisorption mechanisms [128]. $CO_2$ retention by physisorption has also been performed by using aerogels of fibrillated nanocellulose impregnated with acetate-functionalized crystalline nanocellulose suspension [129].

## 6. Conclusions

Nanocellulose can be obtained through increasingly important processes that respect the environment using advanced technologies from agricultural residues. Sustainable application development depends on issues such as raw material selection, extraction methods, product design and life cycle. The uses of nanocellulose in numerous fields are attractive due to its renewable, biocompatible and biodegradable nature.

Pretreatment is an essential step prior to the effective valorization of agricultural lignocellulosic biomass. However, depending on the pretreatment method, it can be costly and originate potential environmental threats. Alkaline pretreatment is the most used for attacking the structure of the residue, eliminating lignin and facilitating the release of cellulose for its subsequent transformation into nanofibers or nanocrystals.

Finally, it is remarkable that more laboratory-scale research studies are needed in order to improve the use of nanocellulose for sustainable applications and its commercial manufacturing adapted to different end-user applications.

**Author Contributions:** Conceptualization, S.M. and A.J.M.; methodology, S.M., S.P. and A.J.M.; software, A.J.M.; validation, A.J.M.; investigation, F.M.-G. and S.P.; resources, F.M.-G., S.P. and A.J.M.; writing—original draft preparation, S.P. and A.J.M.; writing—review and editing, S.M. and A.J.M.; supervision, A.J.M. and M.D.L.R. All authors have read and agreed to the published version of the manuscript.

**Funding:** This research received no external funding.

**Institutional Review Board Statement:** Not applicable.

**Informed Consent Statement:** Not applicable.

**Conflicts of Interest:** The authors declare no conflict of interest.

## Abbreviations

The following abbreviations are used in this manuscript:

| | |
|---|---|
| AFM | Atomic force microscopy |
| AmimCl | 1-allyl-3-methylimidazolium chloride |
| BmimCl | 1-butyl-3-methylimidazolium chloride |
| BNC | Bacterial nanocellulose |
| CBH | Cellobiohydrolases |
| CNC | Cellulose nanocrystals |
| CNF | Cellulose nanofibers |
| DLS | Dynamic light scattering |
| DSC | Differential scanning calorimetry |
| DTA | Differential thermal analysis |
| HIUS | High intensity ultrasonication |
| IL | Ionic liquids |
| LCB | Lignocellulosic biomass |
| LHW | Liquid hot water |
| MCC | Microcrystalline cellulose |
| SANS | Small-angle neutron scattering |
| SEM | Scanning electron microscopy |
| TAED | Tetraacetylethylenediamine |
| TEM | Transmission electron microscopy |
| TEMPO | 2,2,6,6-tetramethylpiperidine-1-oxyl |
| TGA | Thermogravimetric analysis |

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
