# Peer review of "Nanocellulose from Agricultural Wastes: Products and Applications—A Review"

_processes, doi:10.3390/pr9091594_

Round 1

Reviewer 1 Report

It is a very interesting and comprehensive review article on the production of nanocellulose from agricultural wastes. I believe that the article fulfills its role and meets the journal's requirements.

Line 203: I suggest the Authors to extend the information on the use of chemical pretreatment (using the TEMPO reagent) and ultrasonication (https://www.mdpi.com/1996-1944/13/22/5274); 

Line 223: I guess there should be "BNC" instead "BCN";

Table 3: It would be interesting if the Authors could provide values for cellulose efficiency, i.e. cellulose nanocrystals, cellulose nanofibers, etc. (column: "product") obtained with specific parameters;

Line 393: There should be "chitozan" instead of "chitoxan";

Reviewer 2 Report

This review paper is concerned with a highly relevant topic on producing various nanocellulose types and discusses separate stages of isolation, bleaching and final purification of two morphological types of nanocellulose, more specifically cellulose nanofiber (CNF) and cellulose nanocrystal (CNC).  That is what the authors stated in the abstract.

Given the growing number of publications on the synthesis, properties and application of nanocellulose, the authors undertook to tackle a very complicated challenge. Unfortunately, the authors have failed to achieve the objective set in the present manuscript.

Comments and concerns:

  1. The title implies only one type of raw sources of nanocellulose, the agricultural wastes. In that case, the authors were ought to discuss the synthesis of bacterial cellulose from only nutrient media based on pretreatment products of particularly agricultural wastes when describing the processes for bacterial cellulose which is omitted as a morphological type of nanocellulose in the Abstract. The authors should either remove bacterial cellulose from the review or cope with the hardiest task―make a brief overview of at least review papers on this topic.   
  2. The authors mixed up the two notions, “agricultural wastes” and “lignocellulosic biomass”, and included microcrystalline cellulose in Table 1 and omitted bacterial nanocellulose (BNC) in Table 1, though mentioning that type.
  3. What was the reason that the authors narrowed this way, i.e. abridged the synthetic methods for nanocellulose from agricultural wastes, mentioning three pretreatment methods only (see Fig. 3)? In that case, the manuscript title and the Abstract should then be concretized.
  4. The most essential is missing in the manuscript: the rationale for the low yield of nanocellulose from agricultural wastes. Worse than that, the yield is not discussed in a proper manner at all.
  5. The review manuscript attempted to briefly analyze the application fields of nanocellulose. It was done as poor as possible: it is said in Sub. 5.4. Biomedical applications that BNC is used for “biomedical applications”―this is a well-known fact, but at the same time BNC is not always produced from agricultural wastes.
  6. The manuscript includes 117 Refs, among which there are large review papers on nanocellulose for the past decade, but the manuscript lacks analysis of this valuable information.

Reviewer 3 Report

The present paper "Nanocellulose production from agricultural wastes: a review" claims to deal with the process of the nanocellulose. While reading the paper, there are several weaknesses:

  • the paper consists form several sections. While only one section (4. Conditioning, pretreatments and bleaching processes) deals with the production of the nanocellulose. However, cellulose types and application is well discussed. Thereto, it is not clear the focus of the present paper. Mainly Nanocellulose types and its products are discussed.
  • Only several referenced literature sources are from 2021 year.
  • Many typos are in the text. Please use a spelling checker.
  • Biomass processing and nanocellulose production could be discussed in a more strong way. The process schemes and technological images could strongly benefit the present paper's quality.

Round 2

Reviewer 2 Report

The authors have addressed only one concern out of the six raised. The responses to the first two comments are contradictory to each other. The authors deny the existence of review papers on bacterial cellulose produced from agricultural wastes, therefore I cite three references as evidence below.

The authors have not revised the manuscript. The contents of the paper do not reflect the title because the authors believe the terms “agricultural wastes” and “lignocellulosic biomass” are identical, thereby misinforming the readers.  

The authors have not made sense of the cellulose classification, though their reference list contains enough publications with the adequate classification. Table 1 with reference to primary source [14] (Siqueira, G.; Bras, J.; Dufresne, A. Cellulosic bionanocomposites: a review of preparation, properties and applications. Polymers 2010, 2, 728–765. doi:10.3390/polym2040728) misinforms the readers, as this Ref. does not contain any classification.

I ask the authors to figure out the nanocellulose dimensions listed in Table 1 (millimeters, microns or nanometers?). I advise that the authors use a review paper that has already become classical: Klemm, D., Cranston, E. D., Fischer, D., Gama, M., Kedzior, S. A., Kralisch, D., ... & Rauchfuss, F. (2018). Nanocellulose as a natural source for groundbreaking applications in materials science: Today’s state. Materials Today, 21(7), 720-748. https://doi.org/10.1016/j.mattod.2018.02.001.

Reviewer 3 Report

The authors of the review paper "Nanocellulose production from agricultural wastes: a review" claim to discuss the production of nanocellulose. 

Nanocellulose production is claimed, while the production process is still not properly discussed in the present version of the paper.

For the present paper, maybe benefited from the different title. E.g. "review on Nanocellulose from biomass and its products." Or any similar.

Significant enhancements are necessary and advised to be done. Please consider the previous recommendation and review. Briefly, up-to-date publications (2021), process schemes, and other comments. Paper will benefit from it. 

As well, several shortcomings in the text exist - Figure 3 is not mentioned and discussed in the text. Table 1 is not clear. The title is "Nanocellulose dimensions", while there are not mentioned CNC and CNF, also MCC is not nanocellulose. Other inconsistencies in the text also exist.

It is not clear the contribution and difference of the present review paper regarding the many existing similar "nanocellulose reviews".

Considering the above-mentioned comments - "major revision" of the paper is recomended.
